# Facilitating "Omics" for Phenotype Classification Using a User-Friendly AI-Driven Platform: Application in Cancer Prognostics

Uraquitan Lima Filho [1], Tiago Alexandre Pais [2] and Ricardo Jorge Pais [2,3,*]

[1] URA Informatics Ltd., 103 Oxford House Oxford Road, Manchester M1 7ED, UK; ufilho@urainformatics.com
[2] Bioenhancer Systems Ltd., Office 63 182-184 High Street North, East Ham, London E6 2JA, UK; taspais08@gmail.com
[3] Egas Moniz Center for Interdisciplinary Research, Egas Moniz School of Health & Science, 2829-511 Almada, Portugal
[*] Correspondence: rjpais@bioenhancersystems.com

**Abstract:** Precision medicine approaches often rely on complex and integrative analyses of multiple biomarkers from "omics" data to generate insights that can help with either diagnostic, prognostic, or therapeutical decisions. Such insights are often made using machine learning (ML) models that perform sample classification for a particular phenotype (yes/no). Building such models is a challenge and time-consuming, requiring advanced coding skills and mathematical modelling expertise. Artificial intelligence (AI) is a methodological solution that has the potential to facilitate, optimize, and scale model development. In this work, we developed an AI-based, user-friendly, and code-free platform that fully automated the development of predictive models from quantitative "omics" data. Here, we show the application of this tool with the development of cancer survival prognostics models using real-life data from breast, lung, and renal cancer transcriptomes. In comparison to other models, our generated models rendered performances with competitive sensitivities (72–85%), specificities (76–85%), accuracies (75–85%), and Receiver Operating Characteristic curves with superior Areas Under the Curve (ROC-AUC of 77–86%). Further, we reported the associated sets of genes (biomarkers) and their expression patterns that were predictive of cancer survival. Moreover, we made our models available as online tools to generate prognostic predictions based on the gene expressions of the biomarkers. In conclusion, we demonstrated that our tool is a robust, user-friendly solution for developing bespoke predictive tools from "omics" data, which facilitate precision medicine applications to the point-of-care.

**Keywords:** software tools; bioinformatics; cancer prognostics; predictive modelling

## 1. Introduction

Transcriptomics, proteomics, metabolomics, and lipidomics are examples of high-throughput "omics" methodologies often described as precision medicine approaches, which enable the quantitative screening of multiple key biomarkers [1–3]. These methodologies have been often used to characterize human tissue variability and correlation with diseases such as cancer in attempt to find new predictive biomarkers of disease diagnostics, outcomes, and response to therapy [2–4]. Transcriptomics is of particular importance as it is an affordable and accurate gene expression quantification technique [1]. Often, the identified gene expression biomarkers do not have enough predictive power on their own to provide robust insights into the decision making at the point-of-care [5,6]. This has been a persistent problem for cancer prognostics, as the currently used biomarkers still have a low predictive power, explaining from only 25% to 75% of cases [7].

Appling machine learning (ML) modelling frameworks to "omics" data has been considered as a methodological solution for combining biomarkers and improving the predictive capacity of these biomarkers [6,8–10]. Once applying ML to an "omics" dataset

associated with a phenotype outcome, these frameworks find key features (biomarker signatures) that compose a predictive model with a certain predictive power and performance (e.g., sensitivity, specificity, and accuracy) [6,11]. Models, in turn, are able to score a new input of "omics" data with an unknown phenotype/outcome and make a binary phenotype classification (yes/no) [6,8]. However, the technical challenges and limitations associated with the implementation of ML have prevented the full application of its potential to the point-of-care [12]. One critical limitation is associated with the complexity of developing and validating ML algorithms [10,12]. These are hard and time-consuming tasks that require advanced coding skills and mathematical modelling expertise to successfully implement and test supervised learning classification algorithms [12]. Another is choosing the correct ML algorithms (e.g., random forests, neural networks, support vector machines, and regression models) which are suitable for describing the data [8,9]. In addition, the chosen ML algorithm often has numerous tuning parameters for model refinement, which makes it almost humanly impossible to find the best possible model in a reasonable time without a systematic approach.

Automating ML-based model building and validation through artificial intelligence has been proven to be useful for optimal model generation and provides a much faster and more effective route for achieving better-performing models [13,14]. Genetic and evolution inspired AI algorithms have been used in the past for these purposes. Both approaches are inspired by biological mechanisms in nature, such as gene mutation and natural selection rules, to solve optimization problems in a stochastic manner. While genetic algorithms rely on core genetic operators and fixed-length binary representations, evolutionary ML is more flexible due to its wider range of operators and representations. Although both AI frameworks have been demonstrated to render substantial improvements to the quality of ML models, evolutionary algorithms have less of a tendency for bias and provide the means to better tailor the optimization criteria [15]. Meanwhile, evolutionary-inspired AI may also increase the variability of the results and need to be carefully curated for possible artefacts.

TPOT (genetic) and EvA-3 (evolutionary) are two auto ML algorithms that have been applied to the generation of optimal predictive models for early ovarian cancer detection, aneuploidies detection, and cancer prognostics [13,16,17]. Although these algorithms facilitate model development, they are not user-friendly or code-free. Further, current ML algorithms have been rendering poor-performing predictive models for cancer prognostics using transcriptomics [18]. In this work, we developed a novel AI-driven, user-friendly, and code-free web platform for the automated generation of predictive models from "omics" datasets (https://digitalphenomics.com, accessed on 10 August 2023). Here, we applied the tool using an evolution-inspired algorithm for the generation of breast, lung, and renal cancer survival prognostics models.

## 2. Materials and Methods

### 2.1. Tumour Transcriptomics Datasets

Tumour transcriptomics datasets were built using real-life biomedical data consisting of the TCGA transcriptomics data of tumour biopsies from patients with breast, lung, and renal cancers [19,20]. The transcriptomics data were collected from the 2021 updated records of the Human Protein Atlas database, which contained the mRNA expression of 200 genes from 1075 anonymized cancer patients [21,22]. The collected transcriptomics data were already pre-processed and normalized to express the number of fragments per Kilobase of transcript per million of fragments mapped. We curated the collected data in the same way as previously performed to make it comparable with the previously generated model performances using TPOT [18]. Therefore, we selected the same 58 genes and considered the key components of the signalling pathways involved in the regulation of the epithelial-to-mesenchymal transition, which has a role in cancer invasion and metastasis acquisition [23]. From the patients' collected metadata, we selected the transcriptomes associated with the patients that had been reported to survive for over 5 years after the diagnostics (good prognostics), or those with a reported death occurring in less than 2 years

(poor prognostics). The sample numbers of the datasets are summarized in Table 1. A CSV dataset file for each cancer type was created with the gene IDs (first column) and respective FPKM mRNA expression values of all the patient samples (following columns). We also created a metadata CSV file that mapped the survival phenotype of each patient sample with the expression data on the dataset. The datasets were made available in the digital phenomics platform (https://digitalphenomics.com, accessed on 30 May 2023).

**Table 1.** Cancer transcriptomics datasets and their sampling numbers.

| TCGA Refs | N Poor Prognostics | N Good Prognostics | N Total | Tumour Tissue | Dataset ID |
|---|---|---|---|---|---|
| BRCA | 40 | 199 | 239 | Breast | TCGA BCSD |
| LUSC, LUAD | 231 | 94 | 325 | Lung | TCGA LCSD |
| KICH, KIRC, KIRP | 108 | 210 | 318 | Renal | TCGA RCSD |

*2.2. Platform Development*

The Digital Phenomics Platform version 1.0 was developed under a micro-services architecture design for scaling with a robust performance on multiple servers. These micro-services included: cybersecurity; an encrypted relational database; encrypted models and datasets storage; a private and secure user environment; container systems for independently running microservices (Docker); a queueing system; AI-driven model building; an FTP system; API management and supervision; and visualization tools. Multiple coding languages and frameworks were used for the development of the platform. These included Javascript, Python, HTML, PHP, bash, and Nodejs.

*2.3. Model Generation*

Predictive models were generated on the digital phenomics platform (https://digitalphenomics.com, accessed on 25 June 2023) version 1.0. The model generation used the AI software O2P-Mgen version 1.0 developed by the Bioenhancer Systems LTD. This AI was programmed to conduct all the model training, optimization, refinement, and validation automatedly. Using this tool, the data for the model training were selected by the AI with a proportion that was always lower than 50% of the dataset, leaving the remaining data for testing. The AI performed supervised ML to develop models using an evolution-inspired algorithm that found the best combination of biomarkers under a multi-objective fitness function for the optimal sensitivity and specificity (EvA-3 algorithm version 2.0). The algorithm performed time-dependent learning of individual biomarker candidates with statistical significance, with multiple attempts. Next, through generating decision trees, it applied, in a step-wise manner, the multi-objective fitness function for the biomarker $n + 1$ during the model evolution. The multi-objective fitness function was programmed to optimize the sensitivity, specificity, and ROC-AUC metrics, computed under the model validation step. The non-parametric Mawhinney statistical test was used by the AI to compare between groups. The performance metrics were computed by the AI according to the guidelines of Dankers et al. [24].

To build models, the AI was programmed to search for biomarker characteristics that reflected the up-regulations, down-regulations, gene activations, or gene expression inhibitions (e.g., gene knockouts) in the model training groups (positives vs. negatives). For the cancer datasets, we set the good survival prognostics as positives and the poor survival prognostics as negatives. By default, the AI only selected biomarker up-regulations and down-regulations on the training data when the *p*-values were below 0.05, in order to ensure enough statistical significance. In the case of gene activations and inhibitions, the AI was programmed to look for binary expression behaviour in the data, considering a residual degree of tolerance. Models were constructed by the AI using the generic scoring function (Score), where: *Pi* is the absolute distance between the median level of biomarker *i* on the group of positives for the phenotype and the sample value; *Ni* is the absolute distance between the median level of biomarker *i* on the group of positives for the phenotype and the

sample value; *Wi* is the enrichment of biomarker *i* on the group positive for the phenotype; and *n* is the total number of biomarkers in the model.

$$Score = \sum_{i}^{n} \frac{100Wi(-Pi+)}{Pi+}$$

## 3. Results

### 3.1. Digital Phenomics Platform

We developed a novel, user-friendly platform, Digital Phenomics Platform, tailored for the generation of predictive models from "omics" data. We made this platform available online (http://digitalphenomics.com, accessed on 10 August 2023). The platform is organized into modules that address a particular functionality (Figure 1). The GENERATOR module enables us to build datasets (drag and drop) and build predictive models using the uploaded datasets. Building models is straightforward, only requires pressing the add button (+) or edit icon, and type/modify, models' name, description, AI learning time, and maximum false positive rate allowed. Upon saving the request, the AI initializes the model building, which may take from minutes to hours, depending on the amount of learning time requested.

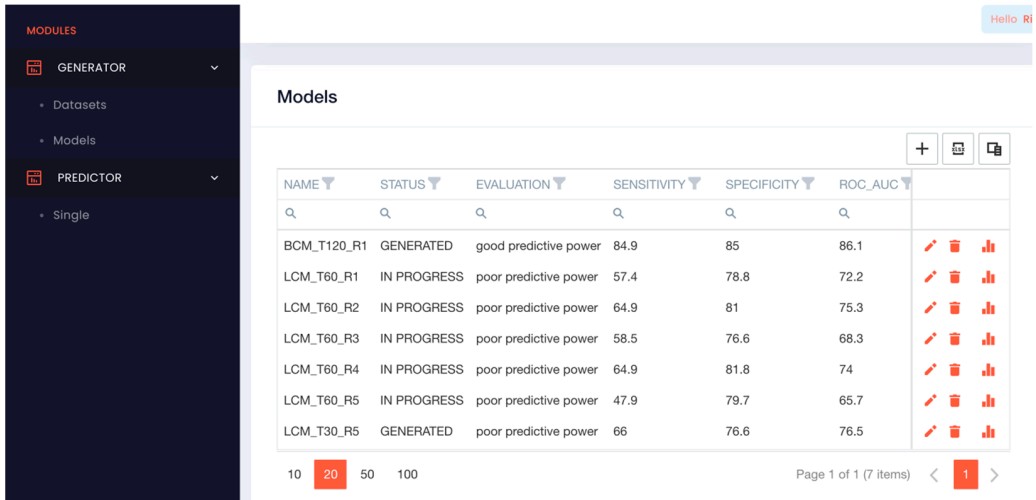

**Figure 1.** Digital Phenomics Platform under the generation of models. On the right, there is an actionable link for the GENERATOR and PREDICTOR modules. Models link on GENERATOR returns a table that shows all models developed and enables editing (pencil icon) or creating a new model (+icon).

Once a model is generated, its characteristics and performance can be analysed by clicking on the bar icon. This generates a table with predictive biomarkers, median levels (positive prediction), the type of predictive regulation (e.g., up-regulation and down-regulation), and the associated *p*-value. A dynamic ROC curve and a heatmap of the predictions are generated, which can optionally be download. The heatmap shows the biomarker scoring, overall predictive score, and outcomes on all the data used to build the model. With this heatmap, users can easily identify false positives and negatives. To generate predictions from unknown samples using the models, we implemented the PREDICT module (Figure 2). In this tool, it is required to insert the values of the model biomarkers and submit them for prediction. Upon submission, the results are shown instantaneously on the platform in a visual and intuitive manner.

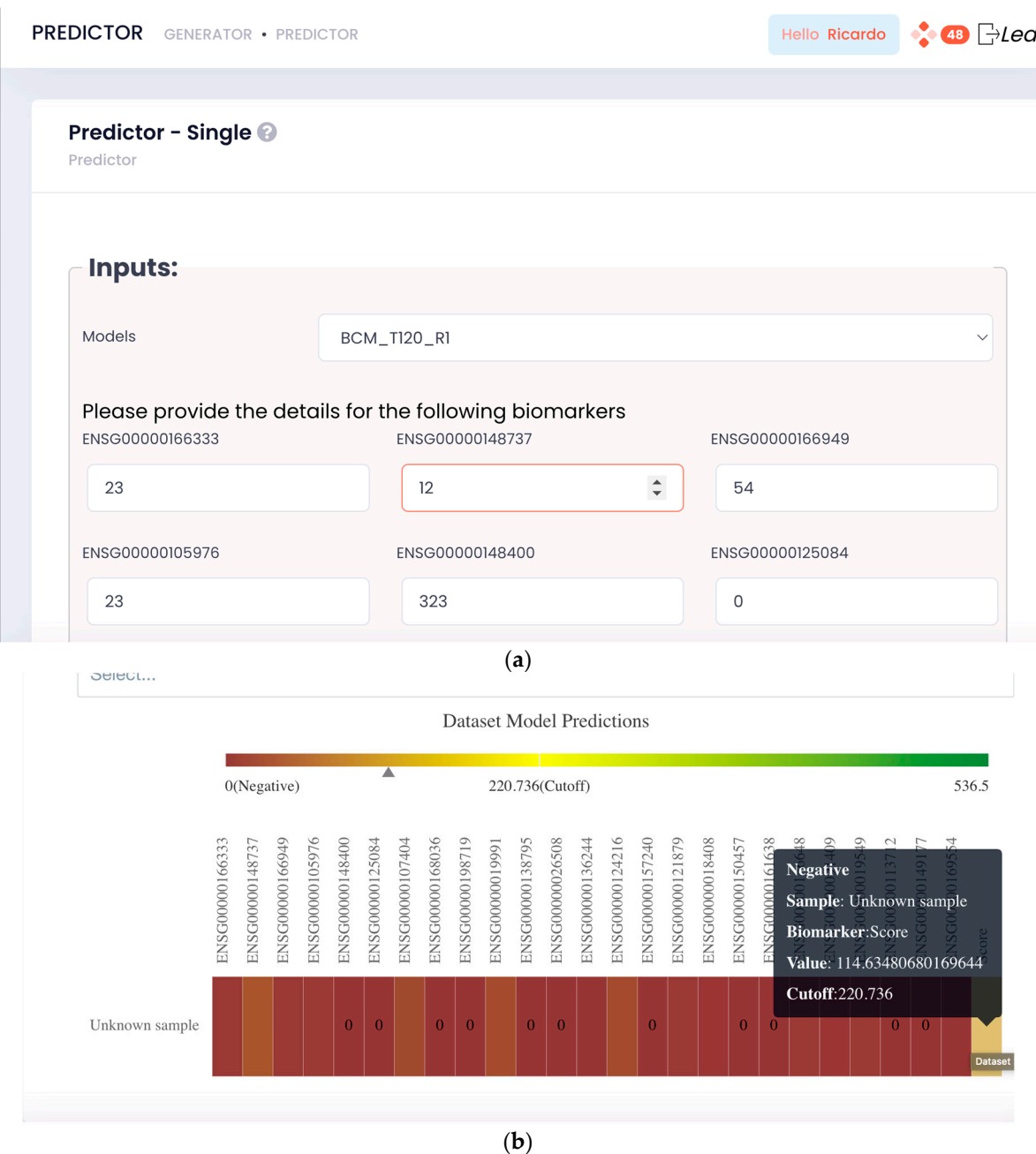

**Figure 2.** The PREDICTOR module tool user interface on the digital phenomics platform. (**a**) Model selection and biomarker input values. (**b**) Unknown sample prediction example using the PRE-DICT tool.

### 3.2. Testing Model Generation with Transcriptomics Data

We tested the model generation potential of the Digital Phenomics Platform with real-life tumour transcriptomics datasets (Table 1). Using these datasets, we requested the generation of 100 models for cancer survival prognostics with AI learning times ranging from 1 to 120 min and a maximum of false positive rate of 25%, repeating it five times. All models were successfully generated with specificities of >75%, fulfilling the user setup requirement and indicating that the tool was robust. The accuracy of the generated prediction tools was also checked by recapitulating the datasets' outcomes and prediction scores, where we manually checked the three models and calculated the sensitivities and specificities. The generated models 'ROCs' AUCs showed an increase in predictive power with learning time, reaching a plateau between 30 and 60 min (Figure 3). The results also

indicated a performance variability in the model building independent of the learning time. On the other hand, these results showed that the predictive power was also dependent on the dataset. In contrast, the number of predictive biomarkers identified by the AI was negatively correlated with the overall models' performance, indicating that the AI was struggling to make models from the renal and lung transcriptomics datasets. Further, the model generation was observed to be approximately 2.4 times the learning time. This was because the AI used the user-defined learning time for the model refinement and required time dedicated to processing the data for finding the synergic effect of the biomarkers' on the model performance.

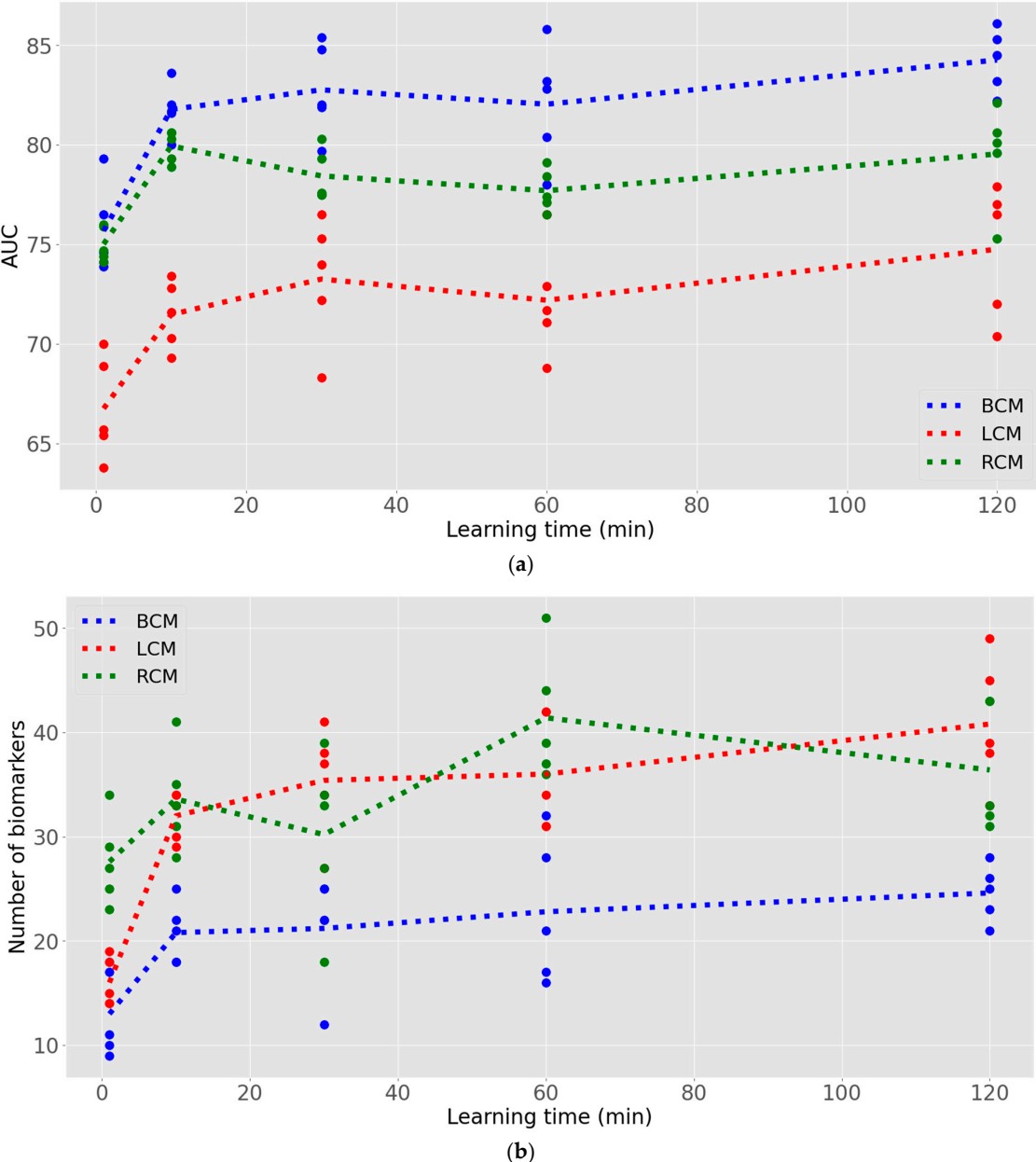

**Figure 3.** Generated cancer survival prognostic models attempt for breast, lung, and renal tumour transcriptomics datasets. (**a**) ROC-AUC as a function of AI learning time. The AUC of the ROC is considered here as the main index of predictive power (**b**) Number of predictive biomarkers in each model as a function of the AI learning time. All models built were conducted at a 25% maximum false positive rate allowed.

### 3.3. Cancer Prognostic Models

We developed breast, lung, and renal cancer survival prognostic models using the transcriptomics datasets (Table 1) and digital phenomics platform. The models were made available for the generation of predictions on https://digitalphenomics.com, accessed on 10 August 2023. The models' performances, the number of predicted biomarkers that composed the models, and their scoring cut-offs are presented in Table 2. We also present in this table the best-performing models generated using TPOT on the same datasets. The obtained models' ROC curves (Figure 4) and their area under the curve values (Table 2) indicated that these models had a good predictive power [24], whereas the models generated with TPOT had an inferior predictive power. The sensitivity, specificity, and accuracy of the breast cancer prognostic model were superior to 85% (Table 2), indicating that this model had a good performance, making it suitable for making predictions on new cancer transcriptomics data [24]. The lung and renal cancer models had lower performances (Table 2), but with sensitivities, specificities, and accuracies always being superior to 70%, indicating that these were reasonable models for generating predictions on new data [24]. As for the TPOT-generated models, performance metrics lower than 70% were always present in any of the models, indicating that our platform algorithm was more robust in these three cases. The biomarkers of the cancer prognostic models and their associated predicted behaviour are presented in Figure 5. We found mainly up-regulations and some down-regulations of the gene expressions in cancer survival phenotypes. Interestingly, these results showed both distinct and conservative gene expression patterns between breast, lung, and renal cancer. We identified eight genes (PI3K, β-catenin, MET, EGF, TCF, LEF, Delta1, and Frizzled) that had up-regulated expressions and 1 (SNAIL1) with down-regulation, conservative across the three cancers types.

**Table 2.** Performances of the cancer survival prognostic models generated by the digital phenomics platform (EvA-3) in comparison to the performances of the optimal models generated using TPOT.

| Cancer | AI | Optimized Predictive Model | ACC | SPE | SEN | AUC |
|---|---|---|---|---|---|---|
| Breast | EvA-3 | 25 biomarkers scoring (cut-off = 221) | 84.9% | 85.0% | 84.9% | 86.1% |
| | TPOT | Multinomial Naïve Bayes Random Forest | * 84.0% | * 58.0% | * 94.0% | * 53% |
| Lung | EvA-3 | 49 biomarkers scoring (cut-off = 690) | 75.1% | 76.2% | 72.3% | 77.0% |
| | TPOT | KNeighbours Random Forest | * 52.0% | * 83.0% | * 59.0% | * 48.0% |
| Renal | EvA-3 | 43 biomarkers scoring (cut-off = 530) | 77.0% | 79.6% | 75.7% | 82.1% |
| | TPOT | Normalised Random Forest | * 71.0% | * 66.0% | * 94.0% | * 70.0% |

\* Models performance values taken from Table 2 of Pais et al. 2023 [18]. SEN (sensitivity); SPE (specificity); AUC (Area Under the Curve); and ACC (accuracy).

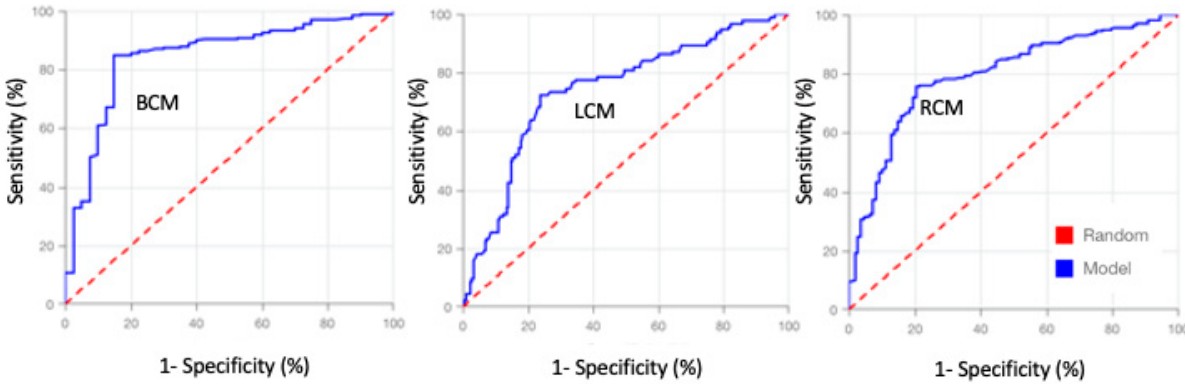

**Figure 4.** Receiver Operating Characteristics (ROC) curves of the selected cancer survival prognostics models (Table 2). ROCs adapted from the ones downloaded from the digital phenomics platform.

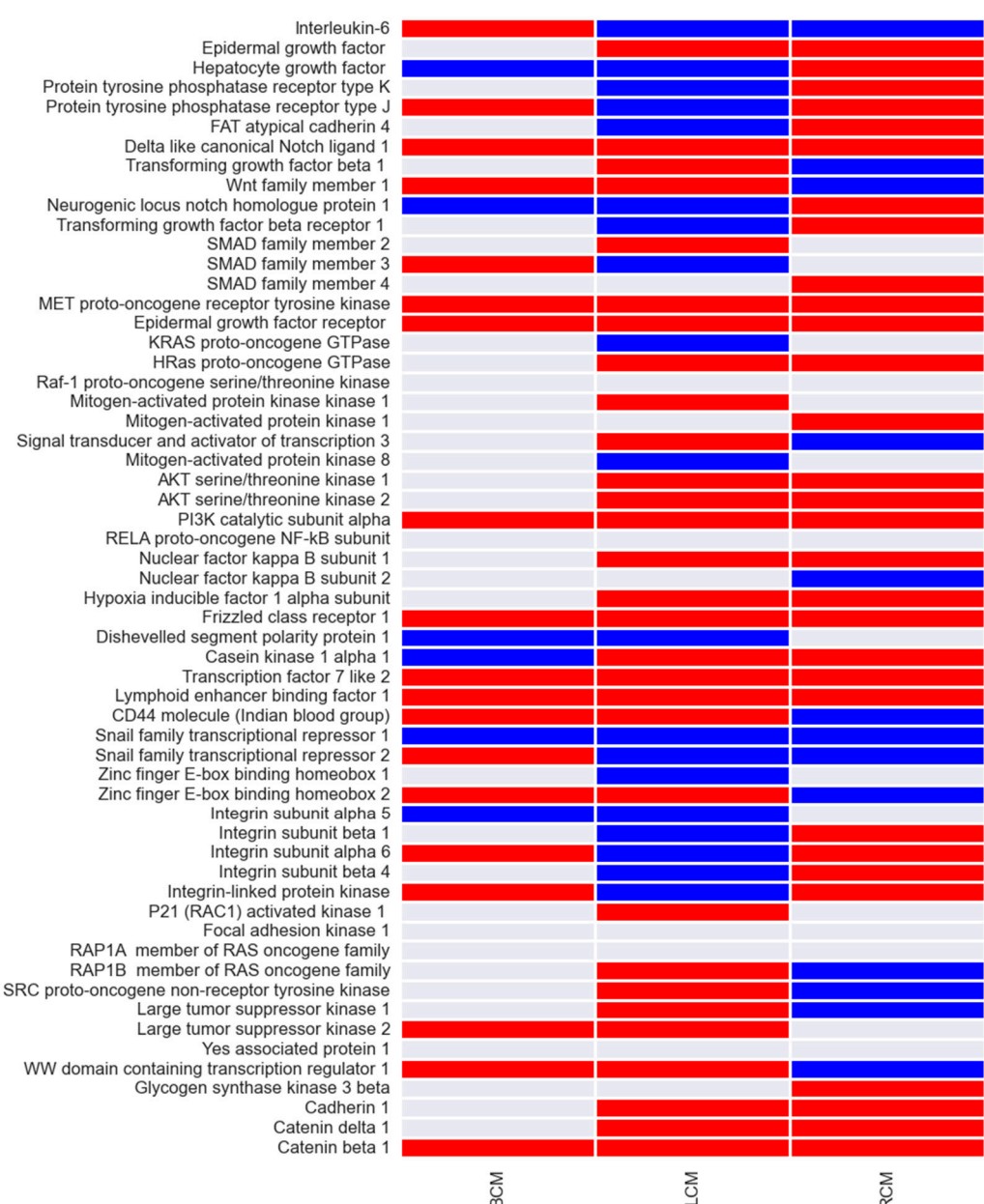

**Figure 5.** The identified regulatory pattern of gene expression of the predictive biomarkers of each prognostic model. Models are indicated by BCM (breast cancer model), LCM (lung cancer model), and RCM (renal cancer model). Red indicates up-regulation, blue indicates down-regulation, and grey indicates no predictive effect. All 58 genes in the datasets are depicted in the y-axis, and gene product names are shown instead of the Ensembl gene IDs.

## 4. Discussion

Current "omics"-based precision medicine frameworks still rely on experienced bioinformaticians with expertise in data science and modelling to generate ML-based predictive [12]. Although this is an ideal scenario for the academy, it is not efficient to implement such frameworks to the point-of-care, as they are required to be scaled to the population level. Our developed solution using AI replaced the role of the specialized bioinformatician modeller and made it possible for unspecialized laboratory personnel to generate and apply predictive models as online tools to the point-of-care. This is mainly because our tool is user-friendly and does not require users to have any coding or advanced mathematical modelling skills. These features are absent from the commonly used machine learning frameworks used in bioinformatics "omics" data analyses, as they are implemented under

the python and R methods and libraries, which, in turn, requires coding and modelling expertise for case-by-case application [12–14]. The tool was designed for a generic and scalable application to large "omics" datasets, where multiple users can work in parallel. This makes it possible for many academic or industry-related laboratories to apply ML to "omics" data without investing in specialized human resources and computational resources, which can be an economic burden above 100,000 GBP per year. Thus, it is expected that our solution may have an impact on opening up new possibilities for academic labs, start-ups, and diagnostic companies that want to focus on precision medicine approaches.

However, the current platform is still limited depending on the quality of the user's data and their capacity to curate, normalize, and process these data before uploading them to the dataset GENERATOR. Future integration with large language model solutions such as OpenAI GTP-4 would potentially improve the dataset generation, enabling users to access and integrate external data and perform AI aid and self-supervised curations. The model generation using our platform brings a distinct AI algorithm based on an evolutionary ML framework in comparison to commonly used ML approaches [15]. The algorithm performs a step-wise identification of biomarkers' up/down-regulations during a model evolution that brings about the advantage of finding synergic effects between $n$ and $n - 1$ identified biomarkers. Further, the implemented multi-objective fitness function enables models to evolve towards more robust predictive tools, not only optimizing towards a particular metric. Instead, models are being evolved towards the best synergic combination in a particular branch of a decision tree, evaluated with multiple metrics (sensitivity, specificity, and ROC-AUC). This has an advantage over the robust auto ML framework TPOT, which only enables rigid one-metric optimisation [13,14]. Another advantage of the tool is that users can also tailor the maximum false positive rate achievable, a feature not commonly enabled in most auto ML tools.

Although our platform solution showed robust results, we highlight some of its limitations and disadvantages in comparison to other solutions. One is the fact that the digital phenomics platform relies only on a scoring-based evolutionary algorithm for model generation, whereas other AI-driven auto ML frameworks, such as TPOT, use a library of algorithms that include random forests, support vector machines, and neural networks [13]. Another limitation is that it can only develop supervised learning classifiers (yes/no), which require a list of categorical features (biomarkers) associated with numerical values (quantitative data). This brings about some uncertainty in the generated predictions when it is near the scoring cut-off of the phenotype yes/no decision. The observed variability associated with the performance of generated models in each attempt and the dependency of the dataset can be also considered as limitations of this technology. This implies trial-and-error attempts from the users to obtain the best-performing model. Furthermore, once new data come, a model is not updated automatedly by the AI. A user intervention to conduct another model development attempt is required. A future version of the AI algorithm should take into account these limitations towards improvements that minimize the impact of these limitations. Perhaps the future integration of multiple AI algorithms, such as TPOT, as users' input choices could enhance the chances of obtaining better models from "omics" data. Further, it would also be important to add AI modules that enable the covering of other types of modelling frameworks such as probabilistic and regression models.

Importantly, the Digital Phenomics Platform rendered promising breast, lung, and renal cancer survival prognostic models from tumour transcriptomics data (Table 2). Our models rendered a much higher predictive power (86% > AUCs > 77%) in comparison to the ones generated using TPOT on the same datasets (70% > AUCs > 48%) [18]. This suggests that our AI-driven modelling framework outcompeted the capacity of the compendium of the ML algorithms implemented in TPOT for transcriptomics data. As we only tested the platform for three real-case transcriptomics datasets, it is still soon to generalise the platform efficacy to other biological systems and even other types of "omics" data, such as proteomics, metabolomics, and lipidomics. Therefore, more studies should be conducted to clarify the potential of the platform AI algorithm as a generic "omics" to phenotype

modelling solution. In comparison to other published ML models, our model, for breast cancer prognostics, performed with a superior sensitivity (86%) in comparison to the reported 35–64%, whereas its specificity was inferior (85%) to the 97–99% [25] urthermore, the obtained AUC for the breast cancer prognostic model (86%) was comparable with the 80–92% reported for other models [25] or the lung and renal cancer prognostic models, we obtained in this work slightly superior performances (up to 10%) in comparison to the ones published using other modelling approaches [26,27]. This suggests that our cancer prognostic models are competitive alternatives to the ones already published. On the other hand, most published models are not applied in clinical decision making, in part because of the difficulty of implementing such ML approaches. As our approach offers the means to easily apply our models with no modelling and coding expertise, we may argue that this work is a step forward towards bringing ML to the point-of-care. Moreover, the models already built from this work may already enhance the current cancer survival prognosis. Now, the next step towards this goal is to make a general practice of the characterization of the tumour biopsy transcriptome in the assessment of cancer prognostics.

Interestingly, the obtained conservative patterns of gene expressions among the cancer types were compatible with the main markers of epithelial phenotype (β-catenin) and incompatible with the markers of the mesenchymal phenotype (SNAIL1) [23,28]. This may partially explain the survival prognostics, as the mesenchymal phenotype and over-expression of SNAIL1 are often correlated with cancer invasion, whereas the epithelial phenotype often correlates with benign cancers [29,30]. However, the other biomarkers identified are considered to be involved in epithelial-to-mesenchymal transitions, known to be hijacked during cancer invasion [31,32]. According to a regulatory network model of epithelial-to-mesenchymal transitions, the identified biomarkers are more compatible with mesenchymal than a more invasive hybrid phenotype [23,33]. Thus, our results agree with this idea, but also highlight the complexity and heterogenicity of cancer deregulations and their correlation with survival prognostics [7,25,27,34]. Moreover, our results from the models may be useful as clues for future studies that aim to understand the molecular mechanisms associated with cancer survival, finding therapeutical targets for specific cancers and therapy evaluation metrics.

## 5. Conclusions

In this work, we developed a novel AI-driven platform for the generation of predictive models from "omics" data. Here, we demonstrated that the platform is a user-friendly, coding-free, robust, and scalable solution suitable to be applied as a precision medicine tool at the point-of-care. This was illustrated with the application of the platform for the generation of breast, lung, and renal cancer prognostics from transcriptomics data. Importantly, with this work, we enabled the usage of competitive and novel cancer prognostic models, which can be accessed online for the generation of predictions through the digital phenomics platform.

**Author Contributions:** Conceptualization, R.J.P. and U.L.F.; methodology, R.J.P. and U.L.F.; software, R.J.P. and U.F; validation, T.A.P.; formal analysis, T.A.P.; data curation, T.A.P.; writing—original draft preparation, U.L.F.; writing—review and editing, R.J.P.; supervision, R.J.P. All authors have read and agreed to the published version of the manuscript.

**Funding:** This research received no external funding.

**Institutional Review Board Statement:** Not applicable.

**Informed Consent Statement:** Not applicable.

**Data Availability Statement:** All data in this research is available at https://digitalphenomics.com, accessed on 10 August 2023.

**Acknowledgments:** We acknowledge Bioenhancer Systems LTD and URA Informatics LTD for supporting the resources necessary to conduct the analysis and maintain the tools online.

**Conflicts of Interest:** U. Filho. and R. Pais. declare a potential conflict of interest as they are directors of Bioenhancer Systems Ltd. and URA Informatics, respectively. T. Pais declares no conflict of interest.

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
