# Peer review of "Facilitating “Omics” for Phenotype Classification Using a User-Friendly AI-Driven Platform: Application in Cancer Prognostics"

_biomedinformatics, doi:10.3390/biomedinformatics3040064_

Round 1

Reviewer 1 Report

Comments and Suggestions for Authors

The manuscript titled "Enabling a User-Friendly AI-Powered Platform for Phenotype Classification from 'Omics' Data: A Case in Cancer Prognostics" presents intriguing content. The paper's notable strengths center around its inventive platform, effective model creation, and its application in the field of cancer prognostics. However, there remains room for improvement in the manuscript.

Minor observations:

  1. It would be beneficial to accentuate the distinctive features and differentiators of this platform, thus enhancing its originality.

  2. The introduction and methods sections involve technical jargon such as "evolution-inspired algorithm." While crucial for experts, providing explanations or definitions tailored to a non-specialist readership would heighten the paper's accessibility.

  3. A deeper exploration within the discussion regarding the potential impact of the generated cancer prognostic models on clinical decision-making is recommended. Expanding on real-world scenarios where these models could be applied would amplify the practical significance of the paper.

  4. Within the results section, offering insights into the biological implications of the identified genes exhibiting consistent expression patterns across various cancer types would add a layer of complexity to the findings.

  5. Although metrics like sensitivity, specificity, accuracy, and AUC are referenced, providing a succinct indication of the platform's precision in achieving accurate predictions could bolster the abstract's potency.

Major points:

  1. A more exhaustive account of the AI algorithm employed for model generation (EvA-3) could enhance the paper's depth.

  2. Despite showcasing the successful development of models and their validation using actual datasets, providing a more comprehensive overview of the validation methodologies or drawing comparisons between the platform's outcomes and those of other established methods, with the inclusion of pertinent examples, is essential.

  3. Delving into potential limitations of the AI-driven platform, such as conceivable biases in training data for models or the challenges tied to extending its utility to different cancer types, would enrich the paper's discourse.

Comments on the Quality of English Language

Minor editing of English language required

Author Response

We thank the reviewer for the comments and suggestions.. Below are the point-by-point answers:

1   “It would be beneficial to accentuate the distinctive features and differentiators of this platform, thus enhancing its originality.”  

REPLY: We have highlighted these features in the first two paragraphs of the discussion.  We further extended these paragraphs to accentuate the distinction between our platform and commonly used methodologies (lines 295-330).    

2 “The introduction and methods sections involve technical jargon such as "evolution-inspired algorithm." While crucial for experts, providing explanations or definitions tailored to a non-specialist readership would heighten the paper's accessibility.”  

REPLY: Agree, we extended introduction (line 70-92) and methods (line 145-152) to improve explanations..    

3   "A deeper exploration within the discussion regarding the potential impact of the generated cancer prognostic models on clinical decision-making is recommended. Expanding on real-world scenarios where these models could be applied would amplify the practical significance of the paper."

REPLY: Agree, we extended discussion paragraph 5  to improve explanations ( line 376-383 ).    

4  "Within the results section, offering insights into the biological implications of the identified genes exhibiting consistent expression patterns across various cancer types would add a layer of complexity to the findings. "

REPLY: we address this point  in the discussion, in particular  paragraph 5 ( line 384-398 ). We improved text for this purpose.      

5 "Although metrics like sensitivity, specificity, accuracy, and AUC are referenced, providing a succinct indication of the platform's precision in achieving accurate predictions could bolster the abstract's potency". 

REPLY: we changed abstract to include some metrics values and reinforce its sound.    

6 "A more exhaustive account of the AI algorithm employed for model generation (EvA-3) could enhance the paper's depth".  

REPLY: We add more detail on the algorithm (lines 145-152). Unfortunately, we cannot add more detail as it is under patent application and commercial protection.    

7 "Despite showcasing the successful development of models and their validation using actual datasets, providing a more comprehensive overview of the validation methodologies or drawing comparisons between the platform's outcomes and those of other established methods, with the inclusion of pertinent examples, is essential."

REPLY: We have compared the generated models using our platform with the models generated using a well established framework for auto ML (TPOT). This was presented in the paragraph 4 of discussion.  We believe this is valuable comparison as the datasets are the exactly the same. Further, the performances are from the best performing model from TPOT, which in turn evaluates multiple algorithms (random forests, NN, Naive Bayes, etc ). To better convey this information, we modified table 2, where we included the TPOT models and their performances. Also adapted text in results (lines 243-258) and discussion (lines 352-383). Also important to mention that we compare our models performance with most recent  published models to access if our platform is able to add new alternatives of impact (lines 369-376).              

8. "Delving into potential limitations of the AI-driven platform, such as conceivable biases in training data for models or the challenges tied to extending its utility to different cancer types, would enrich the paper's discourse."

REPLY: We  improve the text of discussion to account for this issue (lines 314-351).   

Reviewer 2 Report

Comments and Suggestions for Authors

The paper aims to present an AI-powered, user-friendly, code-free platform for developing predictive models from quantitative "omics" data. The study showcases the application of this platform in creating cancer survival prognostic models using data from breast, lung, and renal cancer transcriptomes. The primary strength of the paper is its integration of advanced AI solutions to bridge the gap between complex "omics" data and its practical application in precision medicine. 

Overall, the manuscript is well written, with novelty and significance in content. However, there are still several minor points can be improved. Below are detailed comments:

1. The authors may provide some motivations on why using evolution-inspired algorithm and discuss its Pros and Cons versus other AI/ML algorithm to give readers better understanding of the advantage and limitations of the model.

2. Elaboration on how the input was processed, normalized, and prepared forthe platform might be beneficial.

3. Discussion on how the large language model might contribute to the platform in the future could be interesting to some audience.

4. It might be worthwhile to compare the platform's predictions with other established predictive models for cancer survival to gauge its relative performance.

Author Response

We thank the reviewer for the kind comments and suggestions. Below are the point-by-point answers:

1   "The authors may provide some motivations on why using evolution-inspired algorithm and discuss its Pros and Cons versus other AI/ML algorithm to give readers better understanding of the advantage and limitations of the model."

REPLY:  We add more detail on introduction (lines 70-92) and discussion  (lines 314-350). 

2 "Elaboration on how the input was processed, normalized, and prepared forthe platform might be beneficial. "

REPLY: The tool was not designed for pre-processing of data, instead takes already processed data as  inputs. However, we add more detail on the data was preprocessed  (lines 99-104).

3  "Discussion on how the large language model might contribute to the platform in the future could be interesting to some audience."

REPLY: Interesting idea, as we planning to go down that route. We include as future work in the discussion section (lines 314-318).

4 "It might be worthwhile to compare the platform's predictions with other established predictive models for cancer survival to gauge its relative performance. "

REPLY: we have included in the discussion, now extended a bit more (lines 352-382). See also the new table 2 and results section (lines 243-258) with comparison with well established modelling framework TPOT that generated models on the same datasets.  

Round 2

Reviewer 1 Report

Comments and Suggestions for Authors

Dear Authors,

All my comments are addressed.
Best wishes.

Comments on the Quality of English Language

Minor editing of English language required

Reviewer 2 Report

Comments and Suggestions for Authors

Thanks for the revision, everything looks good now!